# Solid-State Transformations in Inner Coordination Sphere of [Co(NH_3_)_6_][Fe(C_2_O_4_)_3_]∙3H_2_O as a Route to Access Catalytically Active Co-Fe Materials

**DOI:** 10.3390/ma12020221

**Published:** 2019-01-10

**Authors:** Denis P. Domonov, Sophia I. Pechenyuk, Yulia P. Semushina, Kirill V. Yusenko

**Affiliations:** 1I.V. Tananaev Institute of Chemistry and Technology of Rare Elements and Mineral Raw Materials of RAS KSC (ICTREMR), Akademgorodok, 26a, Apatity, Murmansk region 184209, Russia; domonov@chemy.kolasc.net.ru (D.P.D.); semushina@chemy.kolasc.net.ru (Y.P.S.); 2BAM Federal Institute of Materials Research and Testing, Richard-Willstätter Str. 11, D-12489 Berlin, Germany; kirill.yusenko@bam.de

**Keywords:** double complex salts, coordination sphere, catalyst, single-source precursor, thermal decomposition

## Abstract

Thermal decomposition of [Co(NH_3_)_6_][Fe(C_2_O_4_)_3_]∙3H_2_O in argon atmosphere, at a low heating rate (3°/min), and in large amounts of the initial complex (~0.1 mole), has been studied. It was possible to distinguish four decomposition steps upon heating: In the temperature range of 50–100 °C—the loss of crystal water; 100–190 °C—stability region of dehydrated complex; 230–270 °C—the range of stability of intermediate phase with the formula CoFe(NH_3_)_2_(C_2_O_4_)_2_; 270–350 °C—thermal decomposition of the intermediate with the formation of metallic products and further air oxidation with the formation of Co_1.5_Fe_1.5_O_4_. Catalytic properties of thermolysis products were tested in the decomposition reaction of H_2_O_2_ (inactive), oxidation of acetone (average activity), and decomposition of ammonium perchlorate (highly active).

## 1. Introduction

Iron group metals (Fe, Co and Ni) were considered as a base to access of polymetallic catalysts. Their activity and selectivity can be drastically improved by the addition of other transition elements. The active part of these catalytic systems is a disordered, mainly amorphous, oxide-metal nanostructured layer covering a support [1]. The morphology and stability of polymetallic catalysts strongly depends on the preparation routine and the nature of the precursor. Thermal activation, treatment with an alkaline or acid solutions, with further washing or impregnation, may be used to improve catalytic properties of iron-base catalysts [1,2].

Double complex salts (DCS), containing coordination cation and coordination anion, have been proposed as promising, single-source precursors for polymetallic and oxide materials, with promising catalytic activity [3]. Previously we investigated double complex salts with a general formula [*M*^1^(*A*)_6_][*M*^2^*X*_6_], where *M*^1^, *M*^2^ = Fe, Co, Ni, Cu; *A* = NH_3_, en/2, tn/2, ur (En—diaminoethane; Tn—1,3-diaminopropane; Ur—urea), *X* = CN^−^, C_2_O_4_^2−^/2, as precursors of bimetallic catalysts [4,5,6,7]. Their thermal decomposition products were tested as catalysts for H_2_O_2_ decomposition [4], gas-phase oxidation of ethanol and acetone [5,6], as well as in the solid-state decomposition of NH_4_ClO_4_ [7]. It has been shown, that thermal decomposition products prepared by thermolysis in a hydrogen flow are active in decomposition of H_2_O_2_. Nevertheless, ethanol oxidation and NH_4_ClO_4_ decomposition can be catalyzed by thermal decomposition products in the air. In the most tested samples of these catalysts, residual carbon was detected. It should be noted that a mechanism of carbon formation in the thermal decomposition of carbon, containing double complex salts in a solid state is still unclear.

The chemical [Co(NH_3_)_6_][Fe(C_2_O_4_)_3_]·3H_2_O is a precursor for a mixed Co-Fe-containing catalyst. It is sufficient due to its relative low thermal stablility and absence of any toxic components as products of its thermal decomposition [8]. Gaseous products can be overall removed from the solid products of its thermal decomposition. The chemical [Co(NH_3_)_6_][Fe(C_2_O_4_)_3_]·3H_2_O is an example of carbon, containing double complex salts, which in some conditions does not form carbon as an admixture to its solid thermal decomposition products. These products do not belong to the most catalytically active series. The chemical [Co(NH_3_)_6_][Fe(C_2_O_4_)_3_]·3H_2_O is a member of a broad family of isostructural and isomeric salts [*M*^1^(NH_3_)_6_][*M*^2^(C_2_O_4_)_3_]·*n*H_2_O with *M*^1^ = Co, Ir; *M*^2^ = Fe, Cr. Salts are poorly soluble in water, insoluble in organic solvents, as well as air and water stable. The crystal structure of isostructural analogous salt [Ir(NH_3_)_6_][Fe(C_2_O_4_)_3_]·3H_2_O (CSD 1865520) (ICSD, Fachinformationszentrum Karlsruhe, B-76344 Eggenstein-Leopoldshafen, Germany; e-mail crysdata@fiz-karlsruhe.de) has been salved and refined using a single-crystal X-ray diffraction. Isoformular salt [Co(NH_3_)_6_][Fe(C_2_O_4_)_3_]·3H_2_O is isostructural and crystallizes in the hexagonal system with *a* = *b* = 12.55(2), *c* = 20.65(5) Å at room temperature (Figure 1). The structure can be considered as a close packing of cationic and anionic hexagonal layers. The island structure consists of isolated octahedral [Co(NH_3_)_6_]^3+^ cations, octahedrally coordinated [Fe(C_2_O_4_)_3_]^3−^. The geometric characteristics of cations and anions in the crystal structure are similar to the values characteristic for other salts such as [*M*^1^(NH_3_)_6_]Cl_3_ or K_3_[*M*^2^(C_2_O_4_)_3_] [9].

As a general trend, thermal decomposition in a hydrogen atmosphere proceeds in one step, and the process of thermal degradation in an inert atmosphere can have several separate stages [10], which is the reason we settled on an inert atmosphere. The thermal decomposition in the inert atmosphere (helium, argon, or nitrogen flow) is a solid-phase process in the inner coordination sphere of cations and anions, i.e., all products of their thermolysis are formed due to the interaction of the central ions with ligands coordinated to them. It is important, that thermal decomposition of double complex salts can be characterized as a ligand exchange and interaction between central metallic ions and ligands. The detailed understanding of thermal analysis curves and volatile thermolysis products should be accompanied by material balance analysis to be able to understand how inner coordination spheres change with heating. Special large-scale experiments are necessary to address whose issues.

Double complex salts with Co and Fe, such as [Co*A*_6_][*M*(C_2_O_4_)_3_]·3(4)H_2_O (*A* = NH_3_, ½en; *M* = Fe, Cr) were investigated in much detail [11,12]. Thermal decomposition of [Co(NH_3_)_6_][Fe(C_2_O_4_)_3_]∙3H_2_O in air results in the formation of an oxide mixture below 275 °C. Thermal decomposition in the inert atmosphere results in a formation of a mixture of oxides and CoFe nano-dispersive alloy. Thermal decomposition in a hydrogen flow results in the formation of pure single-phase CoFe alloy. Water, ammonia, CO, and CO_2_ are the main gaseous products of its thermal decomposition. In the temperature range 250–300 °C, the formation of an intermediate phase has been proposed [11]. Nevertheless, intermediate phases were never isolated and investigated to obtain their exact composition and properties.

Here, we present a detailed study of the thermal decomposition of [Co(NH_3_)_6_][Fe(C_2_O_4_)_3_]·3H_2_O in an inert atmosphere (argon flow). It is a model system for an in-depth study of more complex cases of the formation of carbon-containing single-source precursors for catalytically active systems. A key crystalline intermediate was isolated and studied using thermal analysis, infrared (IR) spectroscopy, and powder X-ray diffraction. Gram-scale experiments allowed us to isolate homogeneous air-stable samples and investigate their catalytic activity in model reactions. The catalytic properties of nano-structured bi-metallic powders, which do not contain carbon, were characterized in oxidation-reduction reactions.

## 2. Materials and Methods

A total of 400 g of [Co(NH_3_)_6_][Fe(C_2_O_4_)_3_]∙3H_2_O was synthesized by mixing aqueous solutions of equivalent amounts of [Co(NH_3_)_6_]Cl_3_ and Na_3_[Fe (C_2_O_4_)_3_]·3H_2_O (both were synthetized according to [13]). Elemental compositions of [Co(NH_3_)_6_][Fe(C_2_O_4_)_3_]∙3H_2_O, as well as intermediate products were confirmed by elemental analysis (C, Co, Fe). For the elemental analysis, the compounds were dissolved in a concentrated HCl water solution. The metal concentrations were further determined by atomic absorption spectrometry using an AAnalyst-400 spectrometer. Carbon was analyzed using an ELTRA CS-2000 C,H,N automatic analyzer. An elemental analysis of [Co(NH_3_)_6_][Fe(C_2_O_4_)_3_]∙3H_2_O, wt %: C—13.5, Co—11.0, Fe—10.4. Calculated, wt %: C—13.46, Co—11.02, Fe—10.44.

A powder X-ray diffraction (PXRD) study was performed using a Shimadzu XRD 6000 diffractometer (room temperature, CuKα radiation, graphite mono-chromator). IR spectra were collected on a Nicolet 6700 FT-IR spectrometer in KBr tablets. Characteristic absorption maxima in the IR spectrum, cm^−1^: 3547, 3469 v (OH), 3301, 3195 v (NH), 1707, 1679 vas (C = O), 1396 vs. (CO) + v (CC), 1323, 899, 854 vs. (CO) + ā (OC = O), 795 ρ (OH2), 532 v (MO) + v (CC), 482 δcyclo + ā (OC = O).

Specific surface areas were measured using a Tristar 3020 unit. Samples for specific surface area measurements (Table 1) were prepared in a tubular quartz reactor inserted into a tubular Nabertherm RT 50-250/11 furnace.

Thermal gravimetric (TG) curves were obtained using a NETZSCH STA 409 PC/PG (7–10 mg powdered samples were heated in argon flow with heating rates of 3, 5 and 10°/min in a temperature range of 20–1000 °C). Corresponding TG curves for [Co(NH_3_)_6_][Fe(C_2_O_4_)_3_]∙3H_2_O and an intermediate sample **I-1** are presented in Figure 2. Scanning electron microscopy images were obtained using a SEM-Leo 420 microscope (Figure 3).

Intermediate products were isolated after heating [Co(NH_3_)_6_][Fe(C_2_O_4_)_3_]∙3H_2_O up to 250 (**I-1**) and 260 °C (**I-2**) in argon flow in a quartz vessel using a THERMODAT 1955 programmable furnace with a heating rate of 3°/min. Products were tempered at final temperature for an hour and cooled naturally during 24 h. White condensate, from a stream of gaseous products, was collected in a vessel with Raschig rings cooled by ice (0 °C), NH_3,_ and CO_2_ were collected in two coupled Drexel flasks with HCl and NaOH water solutions. Details are given in Table 2.

Catalytic tests were performed using 0.1 g thermolysis products **II-1**, **II-2**, **III** and **IV**. The reactions mentioned above were used to test the catalytic properties of the products. The 1-decomposition of hydrogen peroxide occurred in an aqueous solution in the presence of a solid sample [4]. The first order rate constant has been determined using the residual concentration of H_2_O_2_. The 2-gas-phase oxidation of acetone with oxygen in a current of air, the speed of which was determined by the amount of CO_2_ formed per unit time [5,6]. The 3-solid-phase decomposition of ammonium perchlorate with an additive of 1% of the catalyst to be tested, the efficiency of which was determined by lowering the temperature of complete decomposition of NH_4_ClO_4_ [7].

## 3. Results and Discussion

### 3.1. Thermal Decomposition Products

Carbon-containing double complex salts, such as hexacyanoferrate [14], usually form amorphous carbon as a bi-product of their thermal decomposition in an inert atmosphere. Nevertheless, after the thermal decomposition of [Co(NH_3_)_6_][Fe(C_2_O_4_)_3_]∙3H_2_O above 385 °C, in argon flow, less than 1 wt % of carbon could be detected (Table 1). The product of the thermal decomposition of [Co(NH_3_)_6_][Fe(C_2_O_4_)_3_]∙3H_2_O, obtained above 400 °C, might be pyrophoric on air.

Thermal degradation of [Co(NH_3_)_6_][Fe(C_2_O_4_)_3_]∙3H_2_O (Figure 2) already starts below 100 °C, which corresponds to its dehydration (*ca.* 10 wt %). Tubular crystals of initial salt reveal their shape after complete dehydration above 100 °C (Figure 3). Anhydrous product [Co(NH_3_)_6_][Fe(C_2_O_4_)_3_] is stable up to 190 °C. Further decomposition contains two sequential steps at 190–270 and 270–350 °C.

For the preparation of intermediate **I-1** (60 wt % residual), 75.0 g (0.14 mole) [Co(NH_3_)_6_][Fe(C_2_O_4_)_3_]∙3H_2_O was heated up to 250 °C. White crystalline powder was collected from the inner surface of the furnace, connecting hoses and in the ice-cooled condenser. The powder has high solubility in cold water and can be easily completely dissolved in water. According to IR-spectra, condensed white powder consists of a mixture of ammonium carbonate and ammonium oxalate. According to the weight loss, the intermediate **I-1** corresponds to 225 °C on the TG curve of [Co(NH_3_)_6_][Fe(C_2_O_4_)_3_]∙3H_2_O (Figure 2). Repetition experiments at 260 °C (57 wt % residual) correspond to the intermediate **I-2**. Both intermediate samples according to their weight loss might have idealised stoichiometry CoFe(NH_3_)_2_(C_2_O_4_)_2_. Further thermal decomposition of the intermediate **I-1** (Figure 2) results in a formation of Co_1.5_Fe_1.5_O_4_.

IR spectra of **I-1** and **I-2** are very similar and contain several absorption maxima, cm^−1^: 3195 v (NH), 1636, 1655 v_as_ (C = O), 1313, 799 vs. (CO) + ā (OC = O), 496 δ_cycle_ + ā (OC = O). This indicates the presence of ammonia, carbonyl groups, and cyclic oxalate groups in their structure [15]. All absorption bands in the IR-spectra of initial [Co(NH_3_)_6_][Fe(C_2_O_4_)_3_]∙3H_2_O can be associated with terminal bidentate oxalate groups coordinated to Fe(III) ion, as well as ammonia coordinated to Co(III). Absorption bands in the IR-spectra of **I-1** and **I-2** can be associated with frequencies of bridged and cyclic oxalate ligands. For example, bridged tetradentate C_2_O_4_-group contains two IR-active C-O valent bands at 1628 and 1345 cm^−1^ and does not contain lines at 1650 and 1250–1270 cm^−1^, characteristic of terminal chelate oxalate ligands [15].

Intermediate **I-1** is partially soluble in water with the loss of ammonia. After two months on air, **I-1** and **I-2** lose 80 wt % of ammonia. A total of 10.0 g of fresh **I-1** was treated with 150 mL distilled water. Only 1.7 g of **I-1** was dissolved with a formation of water solution of 0.067 g of Co and 0.009 g of Fe. The solid residue after drying has been identified as a mixture of Co(C_2_O_4_)_2_∙2H_2_O and Fe(C_2_O_4_)_2_∙2H_2_O [16,17]. So, the intermediate product CoFe(NH_3_)_2_(C_2_O_4_)_2_ is not water stable and exists in a very narrow temperature interval (230–270 °C).

A total of 20.0 g of **I-1** was calcined at 420 °C in argon flow. A total of 7.5 g of **II-1** (37.5 wt % residual) was obtained. Sample **II-1** is homogeneous, black not pyrophoric powder with specific surface area of about *S* = 11.2 m^2^/g. A calcination of 20.0 g of **I-2** at 385 °C results in a residual weight of 8.5 g (**II-2**, 42.5 wt % residual) with a specific surface area of about *S* = 17.8 m^2^/g (Table 2). The composition of both samples (**II-1** and **II-2**) can be estimated as CoFeO_0.5_ with 80 wt % of metals. **II-1** and **II-2** are crystalline and can be identified as a mixture of CoFe alloy and Co_1.5_Fe_1.5_O_4_ mixed oxide.

To compare the results of thermolysis of DCS and the intermediate, a weight of 65 g (0.1215 mol) of the original DCS was heated to 370 ° C, at a speed of 3 deg/min in the above-described installation with an hourly exposure. When the ampoule was opened, the residue was heated up to more than 100 °C. A residue of 19.95 g (30.7%) was obtained, product III. The specific surface is 12.1 m^2^/g, the sample is completely crystalline. According to X-ray phase analysis (Figure 4), it is a mixture of oxides M_3_O_4_, (Table 2). Thus, the material burned down, and was, apparently, metal. Repetition of this experiment at 385 ° C under conditions: 60 g (11.22 mmol) in argon gave a residue of 29.5% (17.7 g) of product IV. S_sp_ = 16.1 m^2^/g. When opening the furnace chamber, the material also warmed up, but less than III. The results of the analysis of III and IV, see Table 2. Thermal decomposition of the initial [Co(NH_3_)_6_][Fe(C_2_O_4_)_3_]∙3H_2_O and the intermediate CoFe(NH_3_)_2_(C_2_O_4_)_2_ (**I-2**) finished at the same temperatures, but gave different products (**II-2** and **IV**, correspondently). Consequently, the initial complex, under continuous heating, yields a more disperse and pyrophoric metallic material in comparison with isolated intermediate, which forms mainly CoFe.

### 3.2. Insight into the [Co(NH_3_)_6_][Fe(C_2_O_4_)_3_]·3H_2_O Thermal Decomposition

Thermal decomposition of [Co(NH_3_)_6_][Fe(C_2_O_4_)_3_]∙3H_2_O has the 4 main stages including (i) dehydration corresponding to the loss of 3 molecules of crystal water; (ii) substance of anhydrous complex; (iii) simultaneous loss of 4 molecules of ammonia and one coordinated C_2_O_4_^2-^-ligand corresponds to the formation of CoFe(NH_3_)_2_(C_2_O_4_)_2_ as the main crystalline product, and finally (iv) decomposition of the intermediate corresponds to the reduction of the central ions by two residual oxalate groups with a formation of CoFe alloy.

Dehydrated [Co(NH_3_)_6_][Fe(C_2_O_4_)_3_] is stable up to 180 °C. For the first time, polymeric CoFe(NH_3_)_2_(C_2_O_4_)_2_ intermediate was isolated and investigated ex situ. The intermediate exists in a very narrow temperature interval of 230–270 °C, and has porous microstructure (Figure 3), which is probably a result of gas release through the crystal body.

The decomposition of the CoFe(NH_3_)_2_(C_2_O_4_)_2_ intermediate depends on the heating rate. It decomposes 350 °C (heating rate of 3°/min) and at 400 °C (heating rate 10°/min). The intermediate is probably an individual compound with specific polymeric structure.

An idealized scheme for the thermal decomposition of [Co(NH_3_)_6_][Fe(C_2_O_4_)_3_]∙3H_2_O in an inert atmosphere can be summarized as the following:    initial DCS      100–120 °C  anhydrous DCS   180–230 °C
[Co(NH_3_)_6_][Fe(C_2_O_4_)_3_]·3H_2_O → [Co(NH_3_)_6_][Fe(C_2_O_4_)_3_] + 3H_2_O →
intermediate                   270–350°
[CoFe(NH_3_)_2_(C_2_O_4_)_2_] + CO_2_ + 4NH_3_ + (CO) → CoFe(alloy) + 2NH_3_ + 2CO_2_ + (2 CO) CoFe(alloy) + 2O_2_(atmospheric oxygen) → Co_1.5_Fe_1.5_O_4_ (mixed oxide)

Our current ex situ study gives quite a clear understanding of general trends in [Co(NH_3_)_6_][Fe(C_2_O_4_)_3_]∙3H_2_O thermal decomposition. Nevertheless, a detailed in situ study, using powder X-ray diffraction and IR spectroscopy, will give a much more clearer understanding of secondary oxidation, which might occur under air expose. We are planning to report such a comprehensive study soon for the community.

### 3.3. Catalytic Tests

Samples **III** and **IV** were tested as catalysts of the H_2_O_2_ decomposition, which turned out to be inactive in it (the degree of decomposition of H_2_O_2_ is 3–4% for 75 min). In the oxidation reaction of acetone, product **III** showed the observed reaction rate constant at 350 °C of 1.18 × 10^−4^ s^−1^, and product **IV** has 1.77 × 10^−4^ s^−1^, which is characteristic for bimetallic oxide as thermolysis products. It is at the level of the previously studied products of thermolysis of other Co-Fe double complex salts [5,6].

The product II-2 has been tested in a reaction of decomposition of NH_4_ClO_4_. A presence of 1 wt % of II-2 reduces the temperature of complete decomposition by 150 K. That is much better in comparison with other studies where Cu, Cr, Mn oxides were used [18,19,20]. At the same time, it is comparable with ZnCo_2_O_4_ which reduces the above-stated temperature by 162 K [21]. Partially oxidized CoFe alloy shows relatively high activity as the catalyst of full decomposition of ammonium perchlorate.

Previously, we received very high activity values in a reaction of decomposition of H_2_O_2_ (a rate constant to 10^−2^ L s^−1^ g^−1^ [4]). Such systems were obtained as products of DCS thermal decomposition and contain above 10 wt % of residual carbon, and have a specific surface area of about 100 m^2^ g^−1^ [4,5,6,7]. Low specific surfaces areas and the absence of residual carbon can be correlated with a low catalytic activity. Further, the effect of residual carbon on the catalytic activity of thermolysis products should be investigated in more detail, considering the local structure of metal-carbon intermediates obtained by the thermal decomposition of carbon, containing double complex salts such as [Co(en)_3_][Fe(CN)_6_], [Ni(NH_3_)_6_]_3_[Fe(CN)_6_]_2_ and its analogues [4,5,6,7].

## 4. Conclusions

Thermal decomposition of [Co(NH_3_)_6_][Fe(C_2_O_4_)_3_]·3H_2_O in argon flow corresponds to four clearly distinguishable stages: (a) Dehydration of the initial complex; (b) anhydrous [Co(NH_3_)_6_][Fe(C_2_O_4_)_3_] is stable from 100 to 190 °C; (c) further heating results in a formation of CoFe(NH_3_)_2_(C_2_O_4_)_2_ intermediate stable in the temperature interval of 230–270 °C; (d) intermediate phase completely decomposes upon heating below 350 °C.The initial complex, when heated to 370–420 °C, forms a pyrophoric CoFe alloy, which easily forms Co_1.5_Fe_1.5_O_4_ upon air expose and contains <1 wt % of carbon.Under the same conditions, the intermediate forms a mixture of 80 wt % metals and metal oxides.The final products of thermolysis are catalytically inactive in the decomposition reaction of H_2_O_2_, which we explain by the small specific surfaces and the absence of residual carbon.

## Figures and Tables

**Figure 1 materials-12-00221-f001:**
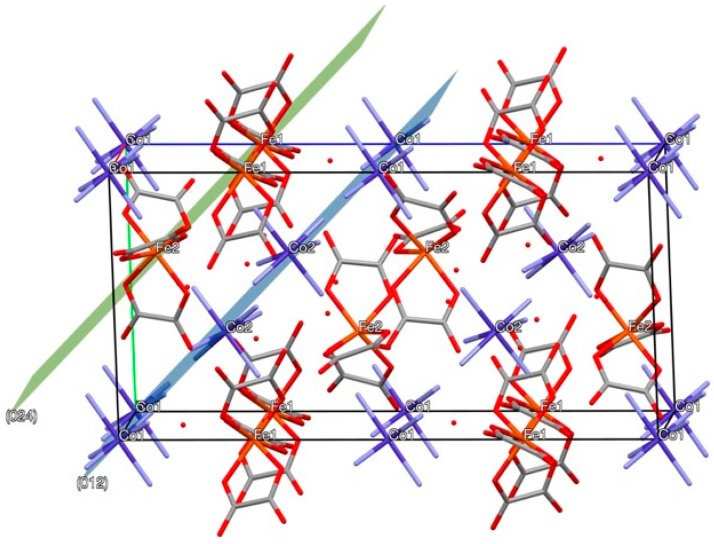
General view of the unit cell in the crystal structure of [Co(NH_3_)_6_][Fe(C_2_O_4_)_3_]∙3H_2_O along the hexagonal axis *a*. Crystallographic planes (0 1 2) and (0 2 4) correspond to [Co(NH_3_)_6_]^3+^ and [Fe(C_2_O_4_)_3_]^3−^∙hexagonal layers.

**Figure 2 materials-12-00221-f002:**
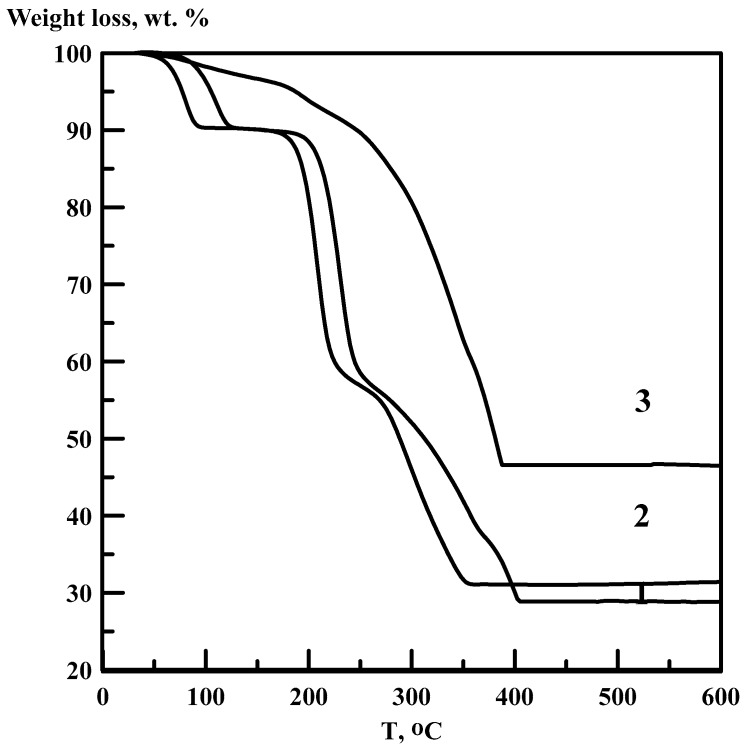
Thermal gravimetric (TG) curves for [Co(NH_3_)_6_][Fe(C_2_O_4_)_3_]∙3H_2_O with heating rate 10°/min (1) 3°/min (2) and TG curve of the intermediate product **I-1** (3) (heating rate 3°/min).

**Figure 3 materials-12-00221-f003:**
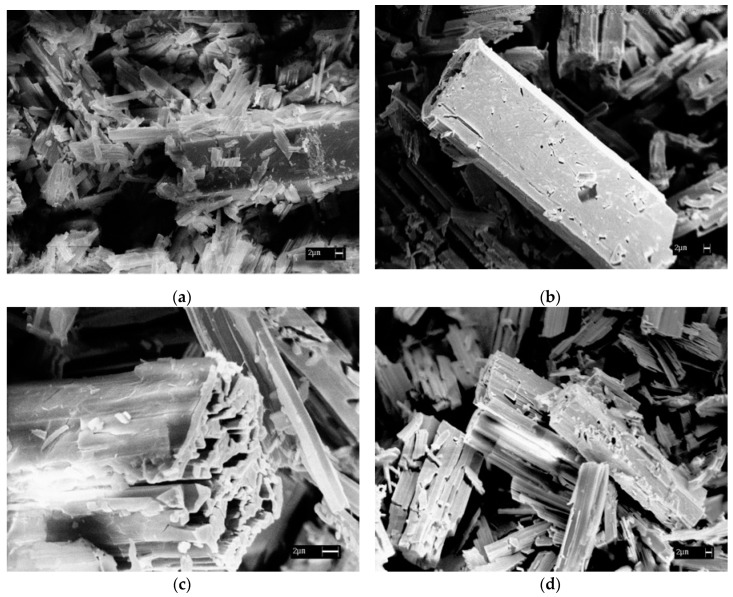
Electronic micrographics of starting [Co(NH_3_)_6_][Fe(C_2_O_4_)_3_]∙3H_2_O single-source precursor (**a**) and intermediate **I-1** (**b**–**d**).

**Figure 4 materials-12-00221-f004:**
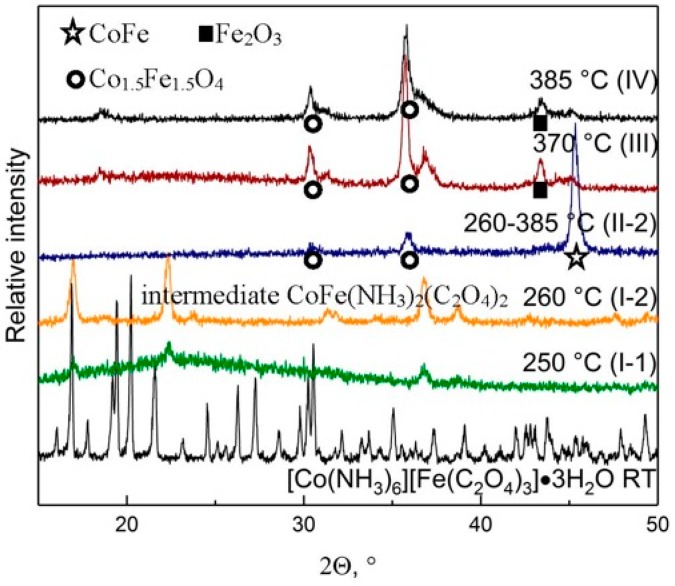
Diffractograms of the precursor salt [Co(NH_3_)_6_][Fe(C_2_O_4_)_3_]∙3H_2_O, intermediates I–III, and final product IV.

**Table 1 materials-12-00221-t001:** Results of thermal treatment of [Co(NH_3_)_6_][Fe(C_2_O_4_)_3_]∙3H_2_O in argon flow (heating rate 5°/min).

Temperature	Residue, wt %	Residual Carbon, wt %	Total Carbon, wt %	S_BET_ m^2^/g	V_∑_, cm^3^/g	Comment
RT	100	13.46	100	-	-	Precursor
300	52.4	13.14	50	29	0.05	tubular crystals
350	41.6	10.3	33	62	0.16	
385	26.1	0.18	<1	45		
400	24.8	0.15	<1	28	0.15	Air stable
400	29.2	0.12	<1	17	0.08	Pyrophoric
450	25.05	0.095	<1	14.5	0.08	Air stable

**Table 2 materials-12-00221-t002:** Results of macroscopic experiments.

Sample	T, °C	Yield, wt %	M.M. Cp.	Content of Elements, at.%			The Released Gas Products, mol/mol DCC		S_BET_ m^2^/g	Composition and Properties
				Co	Fe	C	NH_3_	CO_2_		
**I-1**	250	60.0	321	18.4	17.0	14.9	3.6	0.78	16.8	Light brown, X-ray amorphous Co:Fe:C = 1.03:1:4.08
**I-2**	260	56.9	304	19.7	19.1	14.5	4.1	0.74	36.5	Light brown Co:Fe:C = 0.98:1:3.53
**II-1**	420	37.5	120.4	48.1	45.6	0.06	1.1	1.4	11.2	Black, loose Co:Fe:O = 1:1:0.5
**II-2**	385	42.5	129.2	46.6	46.0	0.14	2.3	2.0	17.8	Black Co:Fe:O = 1:1.04:0.53
**III**	370	30.7 *	164.2	37.4	35.2	0.06	5.4	3.36	12.1	Black, pyrophoric, according to X-ray analysis Co_1.5_Fe_1.5_O_4_ Co:Fe:O = 1:1:2.7
**IV**	385	29.5*	157.7	39.6	38.5	0.07	5.2	2.8	16.1	Black, pyrophoric Co:Fe:O = 1:1.03:1.55

* After air oxidation.

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
