# Peer review of "Solid-State Transformations in Inner Coordination Sphere of [Co(NH3)6][Fe(C2O4)3]∙3H2O as a Route to Access Catalytically Active Co-Fe Materials"

_materials, 2019, doi:10.3390/ma12020221_

Round 1

Reviewer 1 Report

The use of highly abundant first row metals is highly desired and the authors prepare here an interesting complex after thermal decomposition. The paper should be accepted in its present form. One curiosity, have the authors try to dilute the active CoFe catalyst on an inexpensive support such as silica, to decrease the amount (and cost) of metal and increase the performance (higher TON, TOFs), see for instance this recent paper that may be of interest to the autors/readers: MOF-derived metal oxide clusters in porous aluminosilicates: a new catalyst design for the synthesis of bioactive aza-heterocycles (2018) ACS Catalysis, DOI: 10.1021/acscatal.8b03908.

Author Response

Thank you very much for your attention and comments!

In accordance with the recommendations, we tried to improve text. As for the deposition of the active CoFe catalyst on an inexpensive support such as silica, it is not the objective in this paper, but it is planned to study in the future.

Reviewer 2 Report

Although the present manuscript adds some additional information to previous publications from the authors (for example, reference 13), I do not consider that it is enough to be publishable in Materials in its actual form. The authors should emphasize the novelties of this work with relation to previous studies from them and from other authors. It is quite disappointing that more than 60% of the literature cited in the manuscript correspond to self-citations.

If the authors state that intermediate II-2 is a good catalyst for decomposition of ammonium perchlorate, they must provide further information on the experimental results. As well, they have to compare their results with other studies. Some manuscripts on this topic are:

X. Xiao et al., Scientific Reports (2018) 8:7571, DOI: 10.1038/s41598-018-26022-2

I. P. S. Kapoor et al., Propellants Explos. Pyrotech. 2009, 34, 351-356, DOI: 10.1002/prep.200800025

Author Response

Thank you for attention and comments!

We attempted to emphasize the novelties of this work by adding the following sentences to Introduction: «The detailed understanding of thermal analysis curves and volatile thermolysis products should be accompanied by material balance analysis to be able to understand how inner coordination spheres changes with heating. Special large-scale experiments are necessary to address whose issues» and «Here we present detailed study of thermal decomposition of [Co(NH3)6][Fe(C2O4)3]·3H2O in an inert atmosphere (argon flow) as a model system for an in-depth study of more complex cases of formation of carbon-containing single-source precursors for catalytically active systems».

We expanded the references list and reduced the percentage of self-citations.

We compared our results of catalytic activity II-2 with the results of other studies (including the recommended manuscripts).

In addition, we tried to improve the text.

Reviewer 3 Report

Authors have proposed Solid-State transformations in inner coordinaton sphere as a route to access catalytically active  Co-Fe materials. Even though authors have shown some evidences of the proposed four steps via several characterization methods like XRD, TGA and catalytic experiments but the provided evidences do not completely support the proposed hypothesis. 

1) I think authors need to elaborate and explain their hypothesis about why these materials were not catalytically active for H2O2 while they were for acetone and ammonium perchlorate. The present reasoning about absence of residual carbon seems in-sufficient and bit hand wavy. 

2) In addition to ex-situ characterization and evidences, the authors need to support their hypothesis by in-situ characterization because even though the materials are shown to be air-stable but that does not mean no changes to the materials and thus, in-situ work to confirm the hypothesis about transformation in inner coordination sphere is critical.  

Author Response

Thank you very much for your attention and comments!

The presented methods such XRD and TGA prove the existence of 4 stages in the decomposition of the complex, but are not related to the catalytic activity of the product II-2. The amount of data on residual carbon will be substantially complemented by further research. The first three conclusions confirm that transformations in the internal coordination sphere are crucial. We have added a paragraph to the text. «Our current ex situ study gives quite clear understanding of general trends in [Co(NH3)6][Fe(C2O4)3]∙3H2O thermal decomposition. Nevertheless, detailed in situ study using powder X-ray diffraction and IR spectroscopy will give much more clear understanding of secondary oxidation which might occur under air expose. Such comprehensive study we are planning to report soon for the community».

Round 2

Reviewer 2 Report

The authors have highlighted in the revised version the main novelties of their work compared to previous publications. Before being acceptable for publication, the authors must pay attention to the following considerations.

- Page 2. It is not clear in the text if database file CSD 1865520 corresponds to the Ir or Co complex.

- Page 6. Materials II-1 and II-2 are composed of CoFeO2 mixed oxide. However, Fig. 4 gives a Co1.5Fe1.5O4 phase. Why?

- Page 6. Some signals in XRD (Fig. 4) are not assigned. Is there impurities in the samples? Other phases?

- Page 6. In paragraph: “To compare…”, please replace DKS by DCS.

- Page 7. The scheme for the thermal decomposition must be improved and completed, including temperatures, evolution of intermediates and names of the different materials.

Author Response

Thank you for you comments!

- Page 2. It is not clear in the text if database file CSD 1865520 corresponds to the Ir or Co complex.

File CSD 1865520 corresponds to the [Ir(NH3)6][Fe(C2O4)3]•3H2O complex. We changed the text.

- Page 6. Materials II-1 and II-2 are composed of CoFeO2 mixed oxide. However, Fig. 4 gives a Co1.5Fe1.5O4 phase. Why?

Yes, this is a mistake. We corrected the text.

- Page 6. Some signals in XRD (Fig. 4) are not assigned. Is there impurities in the samples? Other phases?

We changed Figure 4.

- Page 6. In paragraph: “To compare…”, please replace DKS by DCS.

Thank you. We replaced DKS by DCS

- Page 7. The scheme for the thermal decomposition must be improved and completed, including temperatures, evolution of intermediates and names of the different materials.

The scheme for the thermal decomposition has been improved and completed

Reviewer 3 Report

Thanks for incorporating some of the suggested changes. We look forward to your report in future with in-situ characterizations.

Author Response

Thank you very much for your comments and helpful advices!